# Maximizing Influence in an Ising Network: A Mean-Field Optimal Solution

**Christopher W. Lynn**
Department of Physics and Astronomy
University of Pennsylvania
chlynn@sas.upenn.edu

**Daniel D. Lee**
Department of Electrical and Systems Engineering
University of Pennsylvania
ddlee@seas.upenn.edu

## Abstract

Influence maximization in social networks has typically been studied in the context of contagion models and irreversible processes. In this paper, we consider an alternate model that treats individual opinions as spins in an Ising system at dynamic equilibrium. We formalize the *Ising influence maximization* problem, which has a natural physical interpretation as maximizing the magnetization given a budget of external magnetic field. Under the mean-field (MF) approximation, we present a gradient ascent algorithm that uses the susceptibility to efficiently calculate local maxima of the magnetization, and we develop a number of sufficient conditions for when the MF magnetization is concave and our algorithm converges to a global optimum. We apply our algorithm on random and real-world networks, demonstrating, remarkably, that the MF optimal external fields (i.e., the external fields which maximize the MF magnetization) shift from focusing on high-degree individuals at high temperatures to focusing on low-degree individuals at low temperatures. We also establish a number of novel results about the structure of steady-states in the ferromagnetic MF Ising model on general graph topologies, which are of independent interest.

## 1 Introduction

With the proliferation of online social networks, the problem of optimally influencing the opinions of individuals in a population has garnered tremendous attention [1–3]. The prevailing paradigm treats marketing as a viral process, whereby the advertiser is given a budget of seed infections and chooses the subset of individuals to infect such that the spread of the ensuing contagion is maximized. The development of algorithmic methods for influence maximization under the viral paradigm has been the subject of vigorous study, resulting in a number of efficient techniques for identifying meaningful marketing strategies in real-world settings [4–6]. While the viral paradigm accurately describes out-of-equilibrium phenomena, such as the introduction of new ideas or products to a system, these models fail to capture reverberant opinion dynamics wherein repeated interactions between individuals in the network give rise to complex macroscopic opinion patterns, as, for example, is the case in the formation of political opinions [7–10]. In this context, rather than maximizing the spread of a viral advertisement, the marketer is interested in optimally shifting the equilibrium opinions of individuals in the network.

To describe complex macroscopic opinion patterns resulting from repeated microscopic interactions, we naturally employ the language of statistical mechanics, treating individual opinions as spins in an Ising system at dynamic equilibrium and modeling marketing as the addition of an external magnetic field. The resulting problem, which we call *Ising influence maximization (IIM)*, has a natural physical interpretation as maximizing the magnetization of an Ising system given a budget of external field. While a number of models have been proposed for describing reverberant opinion dynamics [11], our

use of the Ising model follows a vibrant interdisciplinary literature [12, 13], and is closely related to models in game theory [14, 15] and sociophysics [16, 17]. Furthermore, complex Ising models have found widespread use in machine learning, and our model is formally equivalent to a pair-wise Markov random field or a Boltzmann machine [18–20].

Our main contributions are as follows:

1. We formalize the influence maximization problem in the context of the Ising model, which we call the *Ising influence maximization (IIM)* problem. We also propose the *mean-field Ising influence maximization (MF-IIM)* problem as an approximation to IIM (Section 2).

2. We find sufficient conditions under which the MF-IIM objective is smooth and concave, and we present a gradient ascent algorithm that guarantees an $\epsilon$-approximation to MF-IIM (Section 4).

3. We present numerical simulations that probe the structure and performance of MF optimal marketing strategies. We find that at high temperatures, it is optimal to focus influence on high-degree individuals, while at low temperatures, it is optimal to spread influence among low-degree individuals (Sections 5 and 6).

4. Throughout the paper we present a number of novel results concerning the structure of steady-states in the ferromagnetic MF Ising model on general (weighted, directed) strongly-connected graphs, which are of independent interest. We name two highlights:

   - The well-known pitchfork bifurcation structure for the ferromagnetic MF Ising model on a lattice extends exactly to general strongly-connected graphs, and the critical temperature is equal to the spectral radius of the adjacency matrix (Theorem 3).
   - There can exist at most one stable steady-state with non-negative (non-positive) components, and it is smooth and concave (convex) in the external field (Theorem 4).

## 2  The Ising influence maximization problem

We consider a weighted, directed social network consisting of a set of individuals $N = \{1, \ldots, n\}$, each of which is assigned an opinion $\sigma_i \in \{\pm 1\}$ that captures its current state. By analogy with the Ising model, we refer to $\boldsymbol{\sigma} = (\sigma_i)$ as a spin configuration of the system. Individuals in the network interact via a non-negative weighted coupling matrix $J \in \mathbb{R}_{\geq 0}^{n \times n}$, where $J_{ij} \geq 0$ represents the amount of influence that individual $j$ holds over the opinion of individual $i$, and the non-negativity of $J$ represents the assumption that opinions of neighboring individuals tend to align, known in physics as a ferromagnetic interaction. Each individual also interacts with forces external to the network via an external field $\boldsymbol{h} \in \mathbb{R}^n$. For example, if the spins represent the political opinions of individuals in a social network, then $J_{ij}$ represents the influence that $j$ holds over $i$'s opinion and $h_i$ represents the political bias of node $i$ due to external forces such as campaign advertisements and news articles.

The opinions of individuals in the network evolve according to asynchronous Glauber dynamics. At each time $t$, an individual $i$ is selected uniformly at random and her opinion is updated in response to the external field $\boldsymbol{h}$ and the opinions of others in the network $\boldsymbol{\sigma}(t)$ by sampling from

$$P\left(\sigma_i(t+1) = 1 | \boldsymbol{\sigma}(t)\right) = \frac{e^{\beta\left(\sum_j J_{ij}\sigma_j(t) + h_i\right)}}{\sum_{\sigma_i' = \pm 1} e^{\beta\sigma_i'\left(\sum_j J_{ij}\sigma_j(t) + h_i\right)}}, \tag{1}$$

where $\beta$ is the inverse temperature, which we refer to as the *interaction strength*, and unless otherwise specified, sums are assumed over $N$. Together, the quadruple $(N, J, \boldsymbol{h}, \beta)$ defines our system. We refer to the total expected opinion, $M = \sum_i \langle \sigma_i \rangle$, as the *magnetization*, where $\langle \cdot \rangle$ denotes an average over the dynamics in Eq. (1), and we often consider the magnetization as a function of the external field, denoted $M(\boldsymbol{h})$. Another important concept is the *susceptibility* matrix, $\chi_{ij} = \frac{\partial \langle \sigma_i \rangle}{\partial h_j}$, which quantifies the response of individual $i$ to a change in the external field on node $j$.

We study the problem of maximizing the magnetization of an Ising system with respect to the external field. We assume that an external field $\boldsymbol{h}$ can be added to the system, subject to the constraints $\boldsymbol{h} \geq 0$ and $\sum_i h_i \leq H$, where $H > 0$ is the *external field budget*, and we denote the set of feasible external fields by $\mathcal{F}_H = \{\boldsymbol{h} \in \mathbb{R}^n : \boldsymbol{h} \geq 0, \sum_i h_i = H\}$. In general, we also assume that the system experiences an initial external field $\boldsymbol{b} \in \mathbb{R}^n$, which cannot be controlled.

**Definition 1.** (*Ising influence maximization (IIM)*) Given a system $(N, J, \boldsymbol{b}, \beta)$ and a budget $H$, find a feasible external field $\boldsymbol{h} \in \mathcal{F}_H$ that maximizes the magnetization; that is, find an optimal external field $\boldsymbol{h}^*$ such that

$$\boldsymbol{h}^* = \underset{\boldsymbol{h} \in \mathcal{F}_H}{\arg\max} \, M(\boldsymbol{b} + \boldsymbol{h}). \tag{2}$$

**Notation.** Unless otherwise specified, bold symbols represent column vectors with the appropriate number of components, while non-bold symbols with subscripts represent individual components. We often abuse notation and write relations such as $\boldsymbol{m} \geq 0$ to mean $m_i \geq 0$ for all components $i$.

## 2.1 The mean-field approximation

In general, calculating expectations over the dynamics in Eq. (1) requires Monte-Carlo simulations or other numerical approximation techniques. To make analytic progress, we employ the variational mean-field approximation, which has roots in statistical physics and has long been used to tackle inference problems in Boltzmann machines and Markov random fields [21–24]. The mean-field approximation replaces the intractable task of calculating exact averages over Eq. (1) with the problem of solving the following set of self-consistency equations:

$$m_i = \tanh\left[\beta\left(\sum_j J_{ij} m_j + h_i\right)\right], \tag{3}$$

for all $i \in N$, where $m_i$ approximates $\langle \sigma_i \rangle$. We refer to the right-hand side of Eq. (3) as the *mean-field map*, $\boldsymbol{f}(\boldsymbol{m}) = \tanh\left[\beta(J\boldsymbol{m} + \boldsymbol{h})\right]$, where $\tanh(\cdot)$ is applied component-wise. In this way, a fixed point of the mean-field map is a solution to Eq. (3), which we call a *steady-state*.

In general, there may be many solutions to Eq. (3), and we denote by $\mathcal{M}_{\boldsymbol{h}}$ the set of steady-states for a system $(N, J, \boldsymbol{h}, \beta)$. We say that a steady-state $\boldsymbol{m}$ is *stable* if $\rho(\boldsymbol{f}'(\boldsymbol{m})) < 1$, where $\rho(\cdot)$ denotes the spectral radius and

$$\boldsymbol{f}'(\boldsymbol{m})_{ij} = \left.\frac{\partial f_i}{\partial m_j}\right|_{\boldsymbol{m}} = \beta\left(1 - m_i^2\right) J_{ij} \quad \Rightarrow \quad \boldsymbol{f}'(\boldsymbol{m}) = \beta D(\boldsymbol{m}) J, \tag{4}$$

where $D(\boldsymbol{m})_{ij} = (1 - m_i^2)\delta_{ij}$. Furthermore, under the mean-field approximation, given a stable steady-state $\boldsymbol{m}$, the susceptibility has a particularly nice form:

$$\chi_{ij}^{MF} = \beta\left(1 - m_i^2\right)\left(\sum_k J_{ik}\chi_{kj} + \delta_{ij}\right) \quad \Rightarrow \quad \chi^{MF} = \beta\left(I - \beta D(\boldsymbol{m}) J\right)^{-1} D(\boldsymbol{m}), \tag{5}$$

where $I$ is the $n \times n$ identity matrix.

For the purpose of uniquely defining our objective, we optimistically choose to maximize the maximum magnetization among the set of steady-states, defined by

$$M^{MF}(\boldsymbol{h}) = \max_{\boldsymbol{m} \in \mathcal{M}_{\boldsymbol{h}}} \sum_i m_i(\boldsymbol{h}). \tag{6}$$

We note that the pessimistic framework of maximizing the minimum magnetization yields an equally valid objective. We also note that simply choosing a steady-state to optimize does not yield a well-defined objective since, as $\boldsymbol{h}$ increases, steady-states can pop in and out of existence.

**Definition 2.** (*Mean-field Ising influence maximization (MF-IIM)*) Given a system $(N, J, \boldsymbol{b}, \beta)$ and a budget $H$, find an optimal external field $\boldsymbol{h}^*$ such that

$$\boldsymbol{h}^* = \underset{\boldsymbol{h} \in \mathcal{F}_H}{\arg\max} \, M^{MF}(\boldsymbol{b} + \boldsymbol{h}). \tag{7}$$

## 3 The structure of steady-states in the MF Ising model

Before proceeding further, we must prove an important result concerning the existence and structure of solutions to Eq. (3), for if there exists a system that does not admit a steady-state, then our objective

is ill-defined. Furthermore, if there exists a unique steady-state $\boldsymbol{m}$, then $M^{MF} = \sum_i m_i$, and there is no ambiguity in our choice of objective.

Theorem 3 establishes that every system admits a steady-state and that the well-known pitchfork bifurcation structure for steady-states of the ferromagnetic MF Ising model on a lattice extends exactly to general (weighted, directed) strongly-connected graphs. In particular, for any strongly-connected graph described by $J$, there is a *critical interaction strength* $\beta_c$ below which there exists a unique and stable steady-state. For $\boldsymbol{h} = 0$, as $\beta$ crosses $\beta_c$ from below, two new stable steady-states appear, one with all-positive components and one with all-negative components. Interestingly, the critical interaction strength is equal to the inverse of the spectral radius of $J$, denoted $\beta_c = 1/\rho(J)$.

**Theorem 3.** *Any system $(N, J, \boldsymbol{h}, \beta)$ exhibits a steady-state. Furthermore, if its network is strongly-connected, then, for $\beta < \beta_c$, there exists a unique and stable steady-state. For $\boldsymbol{h} = 0$, as $\beta$ crosses $\beta_c$ from below, the unique steady-state gives rise to two stable steady-states, one with all-positive components and one with all-negative components.*

*Proof sketch.* The existence of a steady-state follows directly by applying Brouwer's fixed-point theorem to $\boldsymbol{f}$. For $\beta < \beta_c$, it can be shown that $\boldsymbol{f}$ is a contraction mapping, and hence admits a unique and stable steady-state by Banach's fixed point theorem. For $\boldsymbol{h} = 0$ and $\beta < \beta_c$, $\boldsymbol{m} = 0$ is the unique steady-state and $\boldsymbol{f}'(\boldsymbol{m}) = \beta J$. Because $J$ is strongly-connected, the Perron-Frobenius theorem guarantees a simple eigenvalue equal to $\rho(J)$ and a corresponding all-positive eigenvector. Thus, when $\beta$ crosses $1/\rho(J)$ from below, the Perron-Frobenius eigenvalue of $\boldsymbol{f}'(\boldsymbol{m})$ crosses 1 from below, giving rise to a supercritical pitchfork bifurcation with two new stable steady-states corresponding to the Perron-Frobenius eigenvector.

*Remark.* Some of our results assume $J$ is strongly-connected in order to use the Perron-Frobenius theorem. We note that this assumption is not restrictive, since any graph can be efficiently decomposed into strongly-connected components on which our results apply independently.

Theorem 3 shows that the objective $M^{MF}(\boldsymbol{b}+\boldsymbol{h})$ is well-defined. Furthermore, for $\beta < \beta_c$, Theorem 3 guarantees a unique and stable steady-state $\boldsymbol{m}$ for all $\boldsymbol{b} + \boldsymbol{h}$. In this case, MF-IIM reduces to maximizing $M^{MF} = \sum_i m_i$, and because $\boldsymbol{m}$ is stable, $M^{MF}(\boldsymbol{b} + \boldsymbol{h})$ is smooth for all $\boldsymbol{h}$ by the implicit function theorem. Thus, for $\beta < \beta_c$, we can use standard gradient ascent techniques to efficiently calculate locally-optimal solutions to MF-IIM. In general, $M^{MF}$ is not necessarily smooth in $\boldsymbol{h}$ since the topological structure of steady-states may change as $\boldsymbol{h}$ varies. However, in the next section we show that if there exists a stable and entry-wise non-negative steady-state, and if $J$ is strongly-connected, then $M^{MF}(\boldsymbol{b}+\boldsymbol{h})$ is both smooth and concave in $\boldsymbol{h}$, regardless of the interaction strength.

# 4  Sufficient conditions for when MF-IIM is concave

We consider conditions for which MF-IIM is smooth and concave, and hence exactly solvable by efficient techniques. The case under consideration is when $J$ is strongly-connected and there exists a stable non-negative steady-state.

**Theorem 4.** *Let $(N, J, \boldsymbol{b}, \beta)$ describe a system with a strongly-connected graph for which there exists a stable non-negative steady-state $\boldsymbol{m}(\boldsymbol{b})$. Then, for any $H$, $M^{MF}(\boldsymbol{b} + \boldsymbol{h}) = \sum_i m_i(\boldsymbol{b} + \boldsymbol{h})$, $M^{MF}(\boldsymbol{b} + \boldsymbol{h})$ is smooth in $\boldsymbol{h}$, and $M^{MF}(\boldsymbol{b} + \boldsymbol{h})$ is concave in $\boldsymbol{h}$ for all $\boldsymbol{h} \in \mathcal{F}_H$.*

*Proof sketch.* Our argument follows in three steps. We first show that $\boldsymbol{m}(\boldsymbol{b})$ is the unique stable non-negative steady-state and that it attains the maximum total opinion among steady-states. This guarantees that $M^{MF}(\boldsymbol{b}) = \sum_i m_i(\boldsymbol{b})$. Furthermore, $\boldsymbol{m}(\boldsymbol{b})$ gives rise to a unique and smooth branch of stable non-negative steady-states for additional $\boldsymbol{h}$, and hence $M^{MF}(\boldsymbol{b} + \boldsymbol{h}) = \sum_i m_i(\boldsymbol{b} + \boldsymbol{h})$ for all $\boldsymbol{h} > 0$. Finally, one can directly show that $M^{MF}(\boldsymbol{b} + \boldsymbol{h})$ is concave in $\boldsymbol{h}$.

*Remark.* By arguments similar to those in Theorem 4, it can be shown that any stable non-positive steady-state is unique, attains the minimum total opinion among steady-states, and is smooth and convex for decreasing $\boldsymbol{h}$.

The above result paints a significantly simplified picture of the MF-IIM problem when $J$ is strongly-connected and there exists a stable non-negative steady-state $\boldsymbol{m}(\boldsymbol{b})$. Given a budget $H$, for any feasible marketing strategy $\boldsymbol{h} \in \mathcal{F}_H$, $\boldsymbol{m}(\boldsymbol{b} + \boldsymbol{h})$ is the unique stable non-negative steady-state, attains the maximum total opinion among steady-states, and is smooth in $\boldsymbol{h}$. Thus, the objective

---

**Algorithm 1:** An $\epsilon$-approximation to MF-IIM

---

**Input:** System $(N, J, \boldsymbol{b}, \beta)$ for which there exists a stable non-negative steady-state, budget $H$,
      accuracy parameter $\epsilon > 0$
**Output:** External field $\boldsymbol{h}$ that approximates a MF optimal external field $\boldsymbol{h}^*$
$t = 0; \boldsymbol{h}(0) \in \mathcal{F}_H; \alpha \in (0, \frac{1}{L})$ ;
**repeat**
   |   $\frac{\partial M^{MF}(\boldsymbol{b}+\boldsymbol{h}(t))}{\partial h_j} = \sum_i \chi_{ij}^{MF}(\boldsymbol{b}+\boldsymbol{h}(t))$;
   |   $\boldsymbol{h}(t+1) = P_{\mathcal{F}_H}\left[\boldsymbol{h}(t) + \alpha \nabla_{\boldsymbol{h}} M^{MF}(\boldsymbol{b}+\boldsymbol{h}(t))\right]$;
   |   $t$++;
**until** $M^{MF}(\boldsymbol{b}+\boldsymbol{h}^*) - M^{MF}(\boldsymbol{b}+\boldsymbol{h}(t)) \leq \epsilon$;
$\boldsymbol{h} = \boldsymbol{h}(t)$;

---

$M^{MF}(\boldsymbol{b}+\boldsymbol{h}) = \sum_i m_i(\boldsymbol{b}+\boldsymbol{h})$ is smooth, allowing us to write down a gradient ascent algorithm that approximates a local maximum. Furthermore, since $M^{MF}(\boldsymbol{b}+\boldsymbol{h})$ is concave in $\boldsymbol{h}$, any local maximum of $M^{MF}$ on $\mathcal{F}_H$ is a global maximum, and we can apply efficient gradient ascent techniques to solve MF-IIM.

Our algorithm, summarized in Algorithm 1, is initialized at a feasible external field. At each iteration, we calculate the susceptibility of the system, namely $\frac{\partial M^{MF}}{\partial h_j} = \sum_i \chi_{ij}^{MF}$, and project this gradient onto $\mathcal{F}_H$ (the projection operator $P_{\mathcal{F}_H}$ is well-defined since $\mathcal{F}_H$ is convex). Stepping along the direction of the projected gradient with step size $\alpha \in (0, \frac{1}{L})$, where $L$ is a Lipschitz constant of $M^{MF}$, Algorithm 1 converges to an $\epsilon$-approximation to MF-IIM in $O(1/\epsilon)$ iterations [25].

## 4.1 Sufficient conditions for the existence of a stable non-negative steady-state

In the previous section we found that MF-IIM is efficiently solvable if there exists a stable non-negative steady-state. While this assumption may seem restrictive, we show, to the contrary, that the appearance of a stable non-negative steady-state is a fairly general phenomenon. We first show, for $J$ strongly-connected, that the existence of a stable non-negative steady-state is robust to increases in $\boldsymbol{h}$ and that the existence of a stable positive steady-state is robust to increases in $\beta$.

**Theorem 5.** *Let $(N, J, \boldsymbol{h}, \beta)$ describe a system with a strongly-connected graph for which there exists a stable non-negative steady-state $\boldsymbol{m}$. If $\boldsymbol{m} \geq 0$, then as $\boldsymbol{h}$ increases, $\boldsymbol{m}$ gives rise to a unique and smooth branch of stable non-negative steady-states. If $\boldsymbol{m} > 0$, then as $\beta$ increases, $\boldsymbol{m}$ gives rise to a unique and smooth branch of stable positive steady-states.*

*Proof sketch.* By the implicit function theorem, any stable steady-state can be locally defined as a function of both $\boldsymbol{h}$ and $\beta$. Using the susceptibility, one can directly show that any stable non-negative steady-state remains stable and non-negative as $\boldsymbol{h}$ increases and that any stable positive steady-state remains stable and positive as $\beta$ increases.

The intuition behind Theorem 5 is that increasing the external field will never destroy a steady-state in which all of the opinions are already non-positive. Furthermore, as the interaction strength increases, each individual reacts more strongly to the positive influence of her neighbors, creating a positive feedback loop that results in an even more positive magnetization. We conclude by showing for $J$ strongly-connected that if $\boldsymbol{h} \geq 0$, then there exists a stable non-negative steady-state.

**Theorem 6.** *Let $(N, J, \boldsymbol{h}, \beta)$ describe any system with a strongly-connected network. If $\boldsymbol{h} \geq 0$, then there exists a stable non-negative steady-state.*

*Proof sketch.* For $\boldsymbol{h} > 0$ and $\beta < \beta_c$, it can be shown that the unique steady-state is positive, and hence Theorem 5 guarantees the result for all $\beta' > \beta$. For $\boldsymbol{h} = 0$, Theorem 3 provides the result.

All together, the results of this section provide a number of sufficient conditions under which MF-IIM is exactly and efficiently solvable by Algorithm 1.

# 5 A shift in the structure of solutions to MF-IIM

The structure of solutions to MF-IIM is of fundamental theoretical and practical interest. We demonstrate, remarkably, that solutions to MF-IIM shift from focusing on nodes of high degree at low interaction strengths to focusing on nodes of low degree at high interaction strengths.

Consider an Ising system described by $(N, J, \boldsymbol{h}, \beta)$ in the limit $\beta \ll \beta_c$. To first-order in $\beta$, the self-consistency equations (3) take the form:

$$\boldsymbol{m} = \beta\left(J\boldsymbol{m} + \boldsymbol{h}\right) \quad \Rightarrow \quad \boldsymbol{m} = \beta(I - \beta J)^{-1}\boldsymbol{h}. \tag{8}$$

Since $\beta < \beta_c$, we have $\rho(\beta J) < 1$, allowing us to expand $(I - \beta J)^{-1}$ in a geometric series:

$$\boldsymbol{m} = \beta\boldsymbol{h} + \beta^2 J\boldsymbol{h} + O(\beta^3) \quad \Rightarrow \quad M^{MF}(\boldsymbol{h}) = \beta \sum_i h_i + \beta^2 \sum_i d_i^{out} h_i + O(\beta^3), \tag{9}$$

where $d_i^{out} = \sum_j J_{ji}$ is the out-degree of node $i$. Thus, for low interaction strengths, the MF magnetization is maximized by focusing the external field on the nodes of highest out-degree in the network, independent of $\boldsymbol{b}$ and $H$.

To study the structure of solutions to MF-IIM at high interaction strengths, we make the simplifying assumptions that $J$ is strongly-connected and $\boldsymbol{b} \geq 0$ so that Theorem 6 guarantees a stable non-negative steady state $\boldsymbol{m}$. For large $\beta$ and an additional external field $\boldsymbol{h} \in \mathcal{F}_H$, $\boldsymbol{m}$ takes the form

$$m_i \approx \tanh\left[\beta\left(\sum_j J_{ij} + b_i + h_i\right)\right] \approx 1 - 2e^{-2\beta(d_i^{in}+b_i+h_i)}, \tag{10}$$

where $d_i^{in} = \sum_j J_{ij}$ is the in-degree of node $i$. Thus, in the high-$\beta$ limit, we have:

$$M^{MF}(\boldsymbol{b} + \boldsymbol{h}) \approx \sum_i \left(1 - 2e^{-2\beta(d_i^{in}+b_i+h_i)}\right) \approx n - 2e^{-2\beta(d_{i*}^{in}+h_{i*}^{(0)}+h_{i*})}, \tag{11}$$

where $i^* = \arg\min_i(d_i^{in} + b_i + h_i)$. Thus, for high interaction strengths, the solutions to MF-IIM for an external field budget $H$ are given by:

$$\boldsymbol{h}^* = \underset{\boldsymbol{h} \in \mathcal{F}_H}{\arg\max}\left(n - 2e^{-2\beta(d_{i*}^{in}+h_{i*}^{(0)}+h_{i*})}\right) \equiv \underset{\boldsymbol{h} \in \mathcal{F}_H}{\arg\max}\,\min_i\left(d_i^{in} + b_i + h_i\right). \tag{12}$$

Eq. (12) reveals that the high-$\beta$ solutions to MF-IIM focus on the nodes for which $d_i^{in} + b_i + h_i$ is smallest. Thus, if $\boldsymbol{b}$ is uniform, the MF magnetization is maximized by focusing the external field on the nodes of smallest in-degree in the network.

We emphasize the strength and novelty of the above results. In the context of reverberant opinion dynamics, the optimal control strategy has a highly non-trivial dependence on the strength of interactions in the system, a feature not captured by viral models. Thus, when controlling a social system, accurately determining the strength of interactions is of critical importance.

# 6 Numerical simulations

We present numerical experiments to probe the structure and performance of MF optimal external fields. We verify that the solutions to MF-IIM undergo a shift from focusing on high-degree nodes at low interaction strengths to focusing on low-degree nodes at high interaction strengths. We also find that for sufficiently high and low interaction strengths, the MF optimal external field achieves the maximum exact magnetization, while admitting performance losses near $\beta_c$. However, even at $\beta_c$, we demonstrate that solutions to MF-IIM significantly outperform common node-selection heuristics based on node degree and centrality.

We first consider an undirected hub-and-spoke network, shown in Figure 1, where $J_{ij} \in \{0, 1\}$ and we set $\boldsymbol{b} = 0$ for simplicity. Since $\boldsymbol{b} \geq 0$, Algorithm 1 is guaranteed to achieve a globally optimal MF magnetization. Furthermore, because the network is small, we can calculate exact solutions to IIM by brute force search. The left plot in Figure 1 compares the average degree of the MF and exact optimal external fields over a range of temperatures for an external field budget $H = 1$, verifying

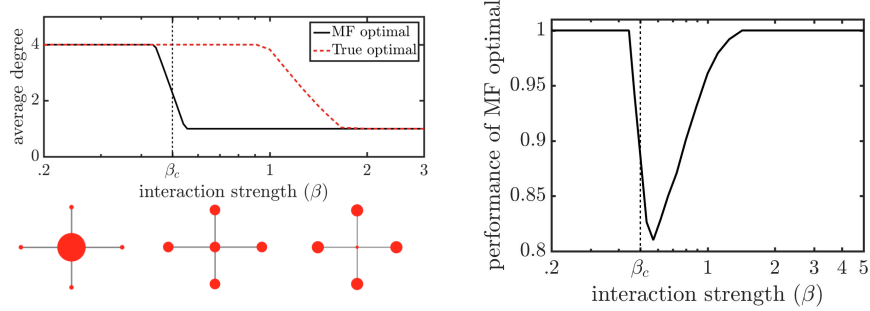

Figure 1: Left: A comparison of the structure of the MF and exact optimal external fields, denoted $\boldsymbol{h}_{MF}^*$ and $\boldsymbol{h}^*$, in a hub-and-spoke network. Right: The relative performance of $\boldsymbol{h}_{MF}^*$ compared to $\boldsymbol{h}^*$; i.e., $M(\boldsymbol{h}_{MF}^*)/M(\boldsymbol{h}_{MF}^*)$, where $M$ denotes the exact magnetization.

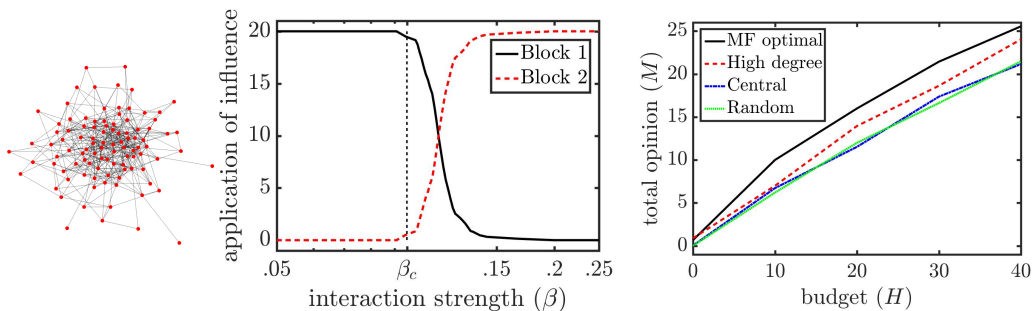

Figure 2: Left: A stochastic block network consisting of a highly-connected community (Block 1) and a sparsely-connected community (Block2). Center: The solution to MF-IIM shifts from focusing on Block 1 to Block 2 as $\beta$ increases. Right: Even at $\beta_c$, the MF solution outperforms common node-selection heuristics.

that the solution to MF-IIM shifts from focusing on high-degree nodes at low interaction strengths to low-degree nodes at high interaction strengths. Furthermore, we find that the shift in the MF optimal external field occurs near the critical interaction strength $\beta_c = .5$. The performance of the MF optimal strategy (measured as the ratio of the magnetization achieved by the MF solution to that achieved by the exact solution) is shown in the right plot in Figure 1. For low and high interaction strengths, the MF optimal external field achieves the maximum magnetization, while near $\beta_c$, it incurs significant performance losses, a phenomenon well-studied in the literature [21].

We now consider a stochastic block network consisting of 100 nodes split into two blocks of 50 nodes each, shown in Figure 2. An undirected edge of weight 1 is placed between each pair of nodes in Block 1 with probability .2, between each pair in Block 2 with probability .05, and between nodes in different blocks with probability .05, resulting in a highly-connected community (Block 1) surrounded by a sparsely-connected community (Block 2). For $\boldsymbol{b} = 0$ and $H = 20$, the center plot in Figure 2 demonstrates that the solution to MF-IIM shifts from focusing on Block 1 at low $\beta$ to focusing on Block 2 at high $\beta$ and that the shift occurs near $\beta_c$. The stochastic block network is sufficiently large that exact calculation of the optimal external fields is infeasible. Thus, we resort to comparing the MF solutions with three node-selection heuristics: one that distributes the budget in amounts proportional to nodes' degrees, one that distributes the budget proportional to nodes' centralities (the inverse of a node's average shortest path length to all other nodes), and one that distributes the budget randomly. The magnetizations are approximated via Monte Carlo simulations of the Glauber dynamics, and we consider the system at $\beta = \beta_c$ to represent the worst-case scenario for the MF optimal external fields. The right plot in Figure 2 shows that, even at $\beta_c$, the solutions to MF-IIM outperform common node-selection heuristics.

We consider a real-world collaboration network (Figure 3) composed of 904 individuals, where each edge is unweighted and represents the co-authorship of a paper on the arXiv [26]. We note that co-authorship networks are known to capture many of the key structural features of social networks

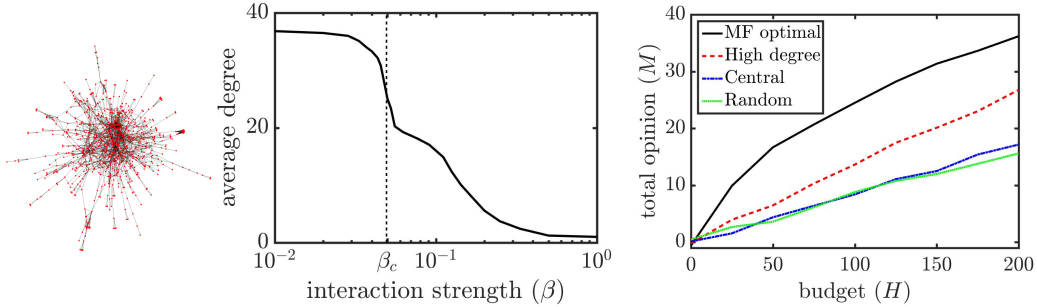

Figure 3: Left: A collaboration network of 904 physicists where each edge represents the co-authorship of a paper on the arXiv. Center: The solution to MF-IIM shifts from high- to low-degree nodes as $\beta$ increases. Right: The MF solution out-performs common node-selection heuristics, even at $\beta_c$.

[27]. For $\boldsymbol{b} = 0$ and $H = 40$, the center plot in Figure 3 illustrates the sharp shift in the solution to MF-IIM at $\beta_c = 0.05$ from high- to low-degree nodes. Furthermore, the right plot in Figure 3 compares the performance of the MF optimal external field with the node-selection heuristics described above, where we again consider the system at $\beta_c$ as a worst-case scenario, demonstrating that Algorithm 1 is scalable and performs well on real-world networks.

# 7   Conclusions

We study influence maximization, one of the fundamental problems in network science, in the context of the Ising model, wherein repeated interactions between individuals give rise to complex macroscopic patterns. The resulting problem, which we call Ising influence maximization, has a natural physical interpretation as maximizing the magnetization of an Ising system given a budget of external magnetic field. Under the mean-field approximation, we develop a number of sufficient conditions for when the problem is concave, and we provide a gradient ascent algorithm that uses the susceptibility to efficiently calculate locally-optimal external fields. Furthermore, we demonstrate that the MF optimal external fields shift from focusing on high-degree individuals at low interaction strengths to focusing on low-degree individuals at high interaction strengths, a phenomenon not observed in viral models. We apply our algorithm on random and real-world networks, numerically demonstrating shifts in the solution structure and showing that our algorithm out-performs common node-selection heuristics.

It would be interesting to study the exact Ising model on an undirected network, in which case the spin statistics are governed by the Boltzmann distribution. Using this elegant steady-state description, one might be able to derive analytic results for the exact IIM problem. Our work establishes a fruitful connection between influence maximization and statistical physics, paving the way for exciting cross-disciplinary research. For example, one could apply advanced mean-field techniques, such as those in [21], to generate efficient algorithms of increasing accuracy. Furthermore, because our model is equivalent to a Boltzmann machine, one could propose a framework for data-based influence maximization based on well-known Boltzmann machine learning techniques.

**Acknowledgements.** We thank Michael Kearns and Eric Horsley for enlightening discussions, and we acknowledge support from the U.S. National Science Foundation, the Air Force Office of Scientific Research, and the Department of Transportation.

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
