[Supplementary Material]

# Supplementary material for:
# Maximizing Influence in an Ising Network:
# A Mean-Field Optimal Solution

**Christopher W. Lynn**
Department of Physics and Astronomy
University of Pennsylvania
chlynn@sas.upenn.edu

**Daniel D. Lee**
Department of Electrical and Systems Engineering
University of Pennsylvania
ddlee@seas.upenn.edu

## 8   Appendix A: Preliminaries

We establish a number of preliminary results that aid the proofs of the theorems in the main text.

### 8.1   Perron-Frobenius

Many of the results in the paper rely on the Perron-Frobenius theorem, which we state here.

**Theorem 7. (Perron-Frobenius)** *Let $J$ be an irreducible non-negative matrix with spectral radius $\rho(J) = r$. Then the following statements hold:*

1. *$J$ has a real, positive, and simple eigenvalue equal to $r$.*
2. *The corresponding eigenvector of $J$ has all-positive components.*
3. *If $0 \leq J \leq A$, for some matrix $A$, then $r_J \leq r_A$.*

It is known that the adjacency matrix of a strongly-connected graph is irreducible, and hence, all of the results of the Perron-Frobenius theorem carry over.

### 8.2   The existence of a unique and stable steady-state for $\beta < \beta_c$

We first show that if our network is strongly-connected, then for $\beta < 1/\rho(J)$, the system exhibits a unique steady-state that is stable under $\boldsymbol{f}$. This result will aid in the proof of Theorem 3 and similar arguments are used in the proof of Theorem 4. The proof in this section is based on Banach's fixed point theorem, but instead of directly showing that $\boldsymbol{f}$ is a contraction mapping on $X = [-1, 1]^n$, we use the spectral properties of $\boldsymbol{f}'$. The following two lemmas relate the contraction mapping property to the spectral radius of $\boldsymbol{f}'$. We note that throughout the proofs, we use the variable $\boldsymbol{x}$ instead of $\boldsymbol{m}$ to indicate a point in $X$ that is not necessarily a steady-state of $\boldsymbol{f}$.

**Lemma 8.** *Let $X$ be a convex subset of Euclidean space and let the function $\boldsymbol{f} : X \to X$ have continuous partial derivatives on $X$. If the Jacobian satisfies*

$$|\boldsymbol{f}'(\boldsymbol{x})| < 1, \tag{13}$$

*for all $\boldsymbol{x} \in X$ and some matrix norm $|\cdot|$, then $\boldsymbol{f}$ satisfies the contraction mapping property on $X$.*

**Lemma 9.** *Given a square matrix $A$ and $\epsilon > 0$, there exists a matrix norm $|\cdot|$ such that*

$$|A| \leq \rho(A) + \epsilon. \tag{14}$$

We are now prepared to show that if $J$ is strongly-connected, then for $\beta < 1/\rho(J)$, $\boldsymbol{f}$ is a contraction mapping on $X$, and hence $\boldsymbol{f}$ admits a unique and stable fixed point on $X$.

**Lemma 10.** *Let $(N, J, \boldsymbol{h}, \beta)$ describe a system with a strongly-connected graph. For $\beta < \beta_c$, there exists a unique and stable steady-state that can be found by iteratively applying $\boldsymbol{f}$ to any point $\boldsymbol{x} \in X$.*

*Proof.* Consider the Jacobian,

$$\boldsymbol{f}'(\boldsymbol{m})_{ij} = \frac{\partial}{\partial m_j} \tanh \left[\beta(J\boldsymbol{m} + \boldsymbol{h})\right]_i = \beta \mathrm{sech}^2 \left[\beta(J\boldsymbol{m} + \boldsymbol{h})\right]_i J_{ij}. \tag{15}$$

Since $|\mathrm{sech}(\cdot)| \leq 1$, $|\boldsymbol{f}'(\boldsymbol{m})_{ij}| \leq \beta J_{ij}$ for all $i, j \in N$. For $\beta < 1/\rho(J)$ and for all $\boldsymbol{m} \in X$, we have

$$\rho(\boldsymbol{f}'(\boldsymbol{m})) \leq \rho(\beta J) = \beta\rho(J) < 1, \tag{16}$$

where the first inequality follows from the Perron-Frobenius theorem and the equality follows from the linearity of $\rho(\cdot)$.

Because the above inequality is strict, there exists an $\epsilon > 0$ such that $\rho(\boldsymbol{f}'(\boldsymbol{x})) + \epsilon < 1$ for all $\boldsymbol{x} \in X$. By Lemma 9 we can choose a matrix norm $|\cdot|$ such that

$$|\boldsymbol{f}'(\boldsymbol{m})| \leq \rho(\boldsymbol{f}'(\boldsymbol{m})) + \epsilon < 1. \tag{17}$$

Since $X$ is a convex subset of Euclidean space, Lemma 8 implies that $\boldsymbol{f}$ satisfies the contraction mapping property on $X$. Since $X$ is a closed and bounded it is also a compact metric space and we can apply Banach's theorem on compact metric spaces to attain the desired result. □

### 8.3 The smoothness of stable non-negative steady-states for increasing $h$

We show that any stable non-negative steady-state $\boldsymbol{m}$ gives rise to a unique and stable branch of steady-states that is smooth and non-decreasing as $\boldsymbol{h}$ increases. We note that this result represents half of the progress toward proving Theorem 5.

**Lemma 11.** *Let $(N, J, \boldsymbol{h}, \beta)$ describe a system with a strongly-connected graph for which there exists a stable steady-state $\boldsymbol{m}$. If $\boldsymbol{m} \geq 0$, then as $\boldsymbol{h}$ increases, $\boldsymbol{m}$ gives rise to a unique and smooth branch of stable non-negative steady-states.*

*Proof.* We first show that any stable steady-state is locally non-decreasing in $\boldsymbol{h}$. Consider the susceptibility from Sec. 2:

$$\chi^{MF} = \beta(I - \beta D(\boldsymbol{m})J)^{-1}D(\boldsymbol{m}) = \beta(I - \boldsymbol{f}'(\boldsymbol{m}))^{-1}D(\boldsymbol{m}). \tag{18}$$

Since $\rho(\boldsymbol{f}'(\boldsymbol{m})) < 1$, Theorem 4.3 of [2] guarantees that the matrix $(I - \boldsymbol{f}'(\boldsymbol{m}))^{-1}$ is non-negative. Furthermore, since $D(\boldsymbol{m})$ is non-negative, we find that $\chi^{MF}$ is non-negative, and hence $\frac{\partial m_i}{\partial h_j} \geq 0$ for all $i, j \in N$.

We now argue that $\boldsymbol{m}$ remains stable as $\boldsymbol{h}$ increases which, by the implicit function theorem, guarantees that $\boldsymbol{m}$ branches uniquely and smoothly. It is sufficient to show that $\rho(\boldsymbol{f}'(\boldsymbol{m})) = \rho(\beta D(\boldsymbol{m})J)$ is non-increasing in $\boldsymbol{h}$. Since $\boldsymbol{m}$ is non-decreasing in $\boldsymbol{h}$, we find that $\boldsymbol{f}'(\boldsymbol{m})$ is entry-wise non-increasing in $\boldsymbol{h}$, and hence Perron-Frobenius guarantees that $\rho(\boldsymbol{f}'(\boldsymbol{m}))$ is non-increasing. □

## 9 Appendix B: Proofs

We now present proofs of the theorems in the main text, noting that we do not present the results in the order that they appear in the text since some earlier results depend on later ones.

### 9.1 Theorem 4

We split Theorem 4 into three separate results. We first show that any stable non-negative steady-state is the unique stable non-negative steady state and that it attains the maximum total opinion among all steady-states. Secondly, we note that Lemma 11 guarantees that any stable non-negative steady-state is smooth and remains stable and non-negative for increasing $\boldsymbol{h}$. Finally, we show that any stable non-negative steady-state is concave in $\boldsymbol{h}$. Together, these results prove Theorem 4.

### 9.1.1 Uniqueness of stable non-negative steady-states

We show that any stable non-negative steady-state is the unique stable non-negative steady-state and achieves the largest total opinion among steady-states. First consider the following lemma.

**Lemma 12.** *Let* $(N, J, \boldsymbol{h}, \beta)$ *describe an arbitrary system and consider any point* $\boldsymbol{x} \in X$. *Then*

$$\boldsymbol{f}^\ell(\mathbf{1}) \geq \boldsymbol{f}^\ell(\boldsymbol{x}), \tag{19}$$

*for any positive integer* $\ell$, *where* $\boldsymbol{f}^\ell(\cdot)$ *denotes the* $\ell^{th}$ *iterative application of* $\boldsymbol{f}$ *and* $\mathbf{1}$ *is the vector of ones of length* $n$.

*Proof.* We proceed by induction. The base case is trivially satisfied. For the inductive step, assume $f^\ell(\mathbf{1})_i \geq f^\ell(\boldsymbol{x})_i$ for some $\ell$ and all $i \in N$. Since $J \geq 0$, $\beta \geq 0$ and $\tanh(\cdot)$ is increasing, we have

$$f^{\ell+1}(\mathbf{1})_i = \tanh\left[\beta\left(Jf^\ell(\mathbf{1}) + \boldsymbol{h}\right)\right]_i \geq \tanh\left[\beta\left(Jf^\ell(\boldsymbol{x}) + \boldsymbol{h}\right)\right]_i = f^{\ell+1}(\boldsymbol{x})_i, \tag{20}$$

for all $\boldsymbol{x} \in X$ and all $i \in N$. $\qquad\square$

We now establish the uniqueness of stable non-negative steady-states.

**Lemma 13.** *Let* $(N, J, \boldsymbol{h}, \beta)$ *describe a system with a strongly-connected network for which there exists a stable non-negative steady-state* $\boldsymbol{m}$. *Then* $\boldsymbol{m}$ *is the unique stable steady-state and can be found by iteratively applying* $\boldsymbol{f}$ *to* $\mathbf{1}$.

*Proof.* Assume there exists a stable non-negative steady-state $\boldsymbol{m}$. By Lemma 12,

$$\boldsymbol{f}^\ell(\mathbf{1}) \geq \boldsymbol{f}^\ell(\boldsymbol{m}) = \boldsymbol{m}, \tag{21}$$

for any $\ell$. This indicates that the sequence $\{\boldsymbol{f}^\ell(\mathbf{1})\}$ is contained in the closed region $X_{\boldsymbol{m}} = \{\boldsymbol{x} \in X : \boldsymbol{x} \geq \boldsymbol{m}\}$. By an argument similar to that in the proof of Lemma 11, we have $\rho(\boldsymbol{f}'(\boldsymbol{x})) \leq \rho(\boldsymbol{f}'(\boldsymbol{m})) < 1$ for all $\boldsymbol{x} \in X_{\boldsymbol{m}}$. By an argument similar to that in the proof of Lemma 10, this indicates that $\boldsymbol{f}$ is a contraction mapping on $X_{\boldsymbol{m}}$.

Following the proof in [3], we show that the sequence $\{\boldsymbol{f}^\ell(\mathbf{1})\}$ is Cauchy. Since $|\boldsymbol{f}(\boldsymbol{x}') - \boldsymbol{f}(\boldsymbol{x})| < |\boldsymbol{x}' - \boldsymbol{x}|$ for all $\boldsymbol{x}, \boldsymbol{x}' \in X_{\boldsymbol{m}}$, there exists a number $q \in (0, 1)$ such that $|\boldsymbol{f}(\boldsymbol{x}') - \boldsymbol{f}(\boldsymbol{x})| \leq q|\boldsymbol{x}' - \boldsymbol{x}|$. By the triangle inequality,

$$|\boldsymbol{x}' - \boldsymbol{x}| \leq |\boldsymbol{x}' - \boldsymbol{f}(\boldsymbol{x}')| + q|\boldsymbol{x}' - \boldsymbol{x}| + |\boldsymbol{f}(\boldsymbol{x}) - \boldsymbol{x}|, \tag{22}$$

which yields

$$|\boldsymbol{x}' - \boldsymbol{x}| \leq \frac{|\boldsymbol{f}(\boldsymbol{x}') - \boldsymbol{x}'| + |\boldsymbol{f}(\boldsymbol{x}) - \boldsymbol{x}|}{1 - q}. \tag{23}$$

Replacing $\boldsymbol{x}$ and $\boldsymbol{x}'$ with $\boldsymbol{f}^\ell(\mathbf{1})$ and $\boldsymbol{f}^k(\mathbf{1})$, respectively, we find

$$|\boldsymbol{f}^k(\mathbf{1}) - \boldsymbol{f}^\ell(\mathbf{1})| \leq \frac{|\boldsymbol{f}^{k+1}(\mathbf{1}) - \boldsymbol{f}^k(\mathbf{1})| + |\boldsymbol{f}^{\ell+1}(\mathbf{1}) - \boldsymbol{f}^\ell(\mathbf{1})|}{1 - q} \tag{24}$$

$$\leq \frac{q^k + q^\ell}{1 - q}|\boldsymbol{f}(\mathbf{1}) - \mathbf{1}|.$$

Since $q < 1$, the last expression goes to zero as $\ell, k \to \infty$, proving that $\{\boldsymbol{f}^\ell(\mathbf{1})\}$ is Cauchy and hence converges to a limit $\boldsymbol{m}^* \in X_{\boldsymbol{m}}$. Furthermore, the limit $\boldsymbol{m}^*$ is a fixed point of $\boldsymbol{f}$, and hence a steady-state of the system, since

$$\boldsymbol{m}^* = \lim_{\ell \to \infty} \boldsymbol{f}^\ell(\mathbf{1}) = \lim_{\ell \to \infty} \boldsymbol{f}(\boldsymbol{f}^{\ell-1}(\mathbf{1})) = \boldsymbol{f}\left(\lim_{\ell \to \infty} \boldsymbol{f}^{\ell-1}(\mathbf{1})\right) = \boldsymbol{f}(\boldsymbol{m}^*). \tag{25}$$

Suppose for contradiction that $\boldsymbol{m}^* \neq \boldsymbol{m}$, and consider the line $(1-t)\boldsymbol{m} + t\boldsymbol{m}^*$ between $\boldsymbol{m}$ and $\boldsymbol{m}^*$ for $t \in [0, 1]$. All points along this line lie in $X_{\boldsymbol{m}}$ and hence $\boldsymbol{f}$ is contractive along the line. We have,

$$|\boldsymbol{f}(\boldsymbol{m}^*) - \boldsymbol{f}(\boldsymbol{m})| = \left|\int_{\boldsymbol{m}}^{\boldsymbol{m}^*} \boldsymbol{f}'(\boldsymbol{x}) \cdot d\boldsymbol{x}\right| \tag{26}$$

$$\leq \int_0^1 |\boldsymbol{f}'\left((1-t)\boldsymbol{m} + t\boldsymbol{m}^*\right)| \, |\boldsymbol{m}^* - \boldsymbol{m}| \, dt,$$

where $|\boldsymbol{f}'(\cdot)|$ represents any matrix norm. Because $\boldsymbol{f}$ is contractive along the line, we can choose a matrix norm that is strictly less than 1. Thus,

$$|\boldsymbol{f}(\boldsymbol{m}^*) - \boldsymbol{f}(\boldsymbol{m})| < \int_0^1 |\boldsymbol{m}^* - \boldsymbol{m}|\, dt = |\boldsymbol{m}^* - \boldsymbol{m}|\,, \tag{27}$$

which is a contradiction. Thus $\boldsymbol{m}^* = \boldsymbol{m}$ and the stable non-negative steady-state is unique. Furthermore, this shows that $\{\boldsymbol{f}^\ell(\boldsymbol{1})\}$ converges to $\boldsymbol{m}$. $\qquad\square$

As a corollary, we find that any stable non-negative steady-state attains the maximum total opinion among all steady-states.

**Corollary 14.** *Let $(N, J, \boldsymbol{h}, \beta)$ describe a system for which there exists a stable non-negative steady-state $\boldsymbol{m}$, and let $\boldsymbol{m}'$ be another steady-state. Then $\boldsymbol{m} \geq \boldsymbol{m}'$.*

*Proof.* By Lemmas 12 and 13 we have

$$\boldsymbol{m} = \lim_{\ell \to \infty} \boldsymbol{f}^\ell(\boldsymbol{1}) \geq \lim_{\ell \to \infty} \boldsymbol{f}^\ell(\boldsymbol{m}') = \boldsymbol{m}', \tag{28}$$

for any steady-state $\boldsymbol{m}'$. $\qquad\square$

### 9.1.2 The concavity of stable non-negative steady-states in $h$

We show for $J$ strongly-connected that any stable non-negative steady-state is concave in $\boldsymbol{h}$.

**Lemma 15.** *Let $(N, J, \boldsymbol{h}, \beta)$ describe a system with a strongly-connected graph for which there exists a stable non-negative steady-state $\boldsymbol{m}$. Then $\boldsymbol{m}$ is concave in $\boldsymbol{h}$.*

*Proof.* We want to show that the Hessian of $m_i$ with respect to $\boldsymbol{h}$ is negative semidefinite for all $i \in N$. The Hessian of $m_i$ with respect to $\boldsymbol{h}$ is given by

$$C_{jk}^{(i)} \equiv \frac{\partial^2 m_i}{\partial h_j \partial h_k} = \frac{\partial}{\partial h_k} \chi_{ij}^{MF}. \tag{29}$$

After taking partials and rearranging we are left with

$$C_{jk}^{(i)} = -2 \sum_{\ell \in N} \chi_{j\ell}^{MF\,T} \left( \frac{m_\ell}{(1 - m_\ell^2)^2} \chi_{i\ell}^{MF} \right) \chi_{\ell k}^{MF} = - \sum_{\ell \in N} Z_{j\ell}^{(i)T} Z_{\ell k}^{(i)}, \tag{30}$$

where $Z_{kj}^{(i)} = \chi_{jk}^{MF} \sqrt{\frac{2m_j}{(1-m_j^2)^2} \chi_{ij}^{MF}}$. Since $m_j, \chi_{ij}^{MF} \geq 0$ for all $i, j \in N$, $Z^{(i)}$ is a real matrix. Thus $C^{(i)}$ is negative semidefinite for all $i \in N$. $\qquad\square$

## 9.2 Theorem 5

We show that any stable non-negative steady-state $\boldsymbol{m}$ gives rise to a unique and stable branch of steady-states that is smooth and non-decreasing as $\boldsymbol{h}$ increases. We also show that if $\boldsymbol{m} > 0$, then $\boldsymbol{m}$ gives rise to a unique and stable branch of steady-states that is smooth and non-decreasing as $\beta$ increases. We note that the first result is given by Lemma 11. To prove the second result, we first show that any stable positive steady-state is locally non-decreasing in $\beta$.

**Lemma 16.** *Let $(N, J, \boldsymbol{h}, \beta)$ describe any system for which there exists a stable positive steady-state $\boldsymbol{m}$. Then $\boldsymbol{m}$ is locally non-decreasing in $\beta$.*

*Proof.* We want to show $\frac{dm_i}{d\beta}$ is non-negative for all $i \in N$. By assumption, $\rho(\boldsymbol{f}'(\boldsymbol{m})) < 1$, allowing us to apply the implicit function theorem, giving

$$\frac{dm_i}{d\beta} = \sum_{j \in N} (\delta_{ji} - \boldsymbol{f}'(\boldsymbol{m})_{ji})^{-1} \frac{\partial f_j}{\partial \beta} \tag{31}$$

$$= \sum_{j \in N} (\delta_{ji} - \boldsymbol{f}'(\boldsymbol{m})_{ji})^{-1} \operatorname{sech}^2 \left[\beta(J\boldsymbol{m} + \boldsymbol{h})\right]_j (J\boldsymbol{m} + \boldsymbol{h})_j.$$

In vector form,

$$\frac{d\boldsymbol{m}}{d\beta} = (I - \boldsymbol{f}'(\boldsymbol{m}))^{-1} D(\boldsymbol{m})(J\boldsymbol{m} + \boldsymbol{h}). \tag{32}$$

Theorem 4.3 of [2] guarantees that the matrix $(I - \boldsymbol{f}'(\boldsymbol{m}))^{-1}$ is non-negative and $D(\boldsymbol{m})$ is also non-negative. Because $\boldsymbol{m} = \tanh\left[\beta(J\boldsymbol{m} + \boldsymbol{h})\right] > 0$, we have $J\boldsymbol{m} + \boldsymbol{h} > 0$, and hence Eq. (32) is non-negative. $\qquad\square$

We now complete the proof of Theorem 5, showing that any stable positive steady-state gives rise to a unique and stable branch of steady-states as $\beta$ increases.

*Proof (Theorem 5).* For contradiction, assume that increasing $\beta$ causes $\boldsymbol{m}$ to lose stability. Because the network is strongly-connected, $\boldsymbol{f}'(\boldsymbol{m}) = \beta D(\boldsymbol{m})J$ is also strongly-connected. Thus, Perron-Frobenius guarantees that $\boldsymbol{f}'(\boldsymbol{m})$ has a simple largest eigenvalue equal to its spectral radius. When $\boldsymbol{m}$ loses stability, this simple eigenvalue crosses one from below. By the Crandall-Rabinowitz theorem [1] and the principle of exchange of stability, the crossing of the simple eigenvalue gives rise to two new sable steady-states. However, Lemma 16 guarantees that $\boldsymbol{m}$ remains positive as we increase $\beta$, which necessitates that both of the new stable steady-states are also initially positive, contradicting Theorem 4. Thus, $\boldsymbol{m}$ cannot lose stability as $\beta$ increases, and hence $\boldsymbol{m}$ gives rise to a unique and smooth branch of stable and positive steady-states. $\qquad\square$

### 9.3   Theorem 3

We show that every system exhibits a steady-state and that the well-known pitchfork bifurcation structure for steady-states of the ferromagnetic MF Ising model on a lattice extends exactly to general (weighted, directed) strongly-connected graphs. In particular, for any strongly-connected graph $J$, there is a critical interaction strength $\beta_c = 1/\rho(J)$ below which there exists a unique and stable steady-state. For $\boldsymbol{h} = 0$, as $\beta$ crosses $\beta_c$ from below, two new stable steady-states appear, one with all-positive components and one with all-negative components.

*Proof (Theorem 3).* We first note that the existence of a steady-state is guaranteed for any system by applying Brouwer's fixed point theorem to $\boldsymbol{f}$. Furthermore, Lemma 10 establishes that for $\beta < 1/\rho(J)$, there is a unique and stable steady-state.

In the case $\boldsymbol{h} = 0$, any system has a steady-state at $\boldsymbol{m}^* = 0$, which we refer to as the *trivial steady-state*. Lemma 10 guarantees that $\boldsymbol{m}^*$ is stable and unique for $\beta < \beta_c$. The implicit function theorem guarantees that we can continue to write $\boldsymbol{m}^*$ uniquely as a function of $\beta$ so long as $\rho(\boldsymbol{f}'(\boldsymbol{m}^*)) = \beta\rho(J) < 1$. If our network is strongly-connected, then the Perron-Frobenius theorem guarantees that as we increase $\beta$, an eigenvalue of $\boldsymbol{f}'$ will first cross 1 when $\beta = 1/\rho(J)$. Furthermore, the largest eigenvalue is simple, which, by the Crandall-Rabinowitz theorem [1], guarantees the appearance of two new steady-states. Furthermore, the new solutions locally lie in the subspace spanned by the eigenvector corresponding to the largest eigenvalue of $\boldsymbol{f}'$, which by the Perron-Frobenius theorem has all positive components. Thus, at $\beta = \beta_c$, a branch of steady-states appears, giving rise to an all-positive steady-state and an all-negative steady-state. By the principle of exchange of stability, the new steady-states adopt the stability of the trivial steady-state, while the trivial steady-state becomes unstable. As we continue to increase $\beta$, Theorem 5 guarantees that the positive (negative) steady-state remains positive (negative) and stable. $\qquad\square$

### 9.4   Theorem 6

We conclude by showing for $J$ strongly-connected that if $\boldsymbol{h} \geq 0$, then there exists a stable non-negative steady-state.

*Proof (Theorem 6).* We first consider $\boldsymbol{h} > 0$. Lemma 10 guarantees that for any $\beta < \beta_c$ there exists a unique and stable steady-state $\boldsymbol{m}$ and that iterative application of $\boldsymbol{f}$ to any $\boldsymbol{x} \in X$ converges to $\boldsymbol{m}$. For induction, choose $\boldsymbol{x} = \mathbf{1}$ and assume $\boldsymbol{f}^\ell(\mathbf{1}) > 0$. Then

$$\boldsymbol{f}^{\ell+1}(\mathbf{1}) = \tanh\left[\beta(J\boldsymbol{f}^\ell(\mathbf{1}) + \boldsymbol{h})\right] > 0. \tag{33}$$

Thus, $\boldsymbol{m} = \lim_{\ell\to\infty} \boldsymbol{f}^\ell(\mathbf{1}) > 0$ at $\beta$. By Theorem 5, the unique branch $\boldsymbol{m}(\beta)$ remains stable and positive for all $\beta' > \beta$. To complete the proof, we note that Theorem 3 covers the case $\boldsymbol{h} = 0$. $\qquad\square$