[Reviews · NeurIPS 2016]

Reviewer 1

Summary

The authors consider a model of opinions dynamics, formalized through the Ising model where individuals opinions are spins on the nodes of a given directed network. A marketing strategy is considered that is an external force acting on the individuals opinions. They formalize the Ising Influence Maximization (IIM) problem that is to maximize the expected opinions (under the stationary distribution) at the fixed marketing budget. The authors establish many results: the mean-field solution to the computation of the expected steady-state opinions exists; exact solution to the IIM problem exists under sufficient conditions; optimal marketing strategies exhibit phase transitions. They propose a gradient-ascent algorithm to approximate the solution. Numerical experiments illustrate the theoretical results.

Qualitative Assessment

The paper is quite easy to read. All the technical parts are put in the supplementary material, that contains all the necessary elements for establishing the proofs. Thus the paper is understandable by a large audience.

Confidence in this Review

2-Confident (read it all; understood it all reasonably well)


Reviewer 2

Summary

The authors defined a new influence maximization model based on Ising model to capture opinion dynamics at equilibrium. Using a mean field approximation, they proved that many classic Ising model steady states can also be found on general graphs, leading to an efficient algorithm based on gradient ascent. Finally, empirical tests are used to verify these theoretical results.

Qualitative Assessment

While the Ising influence maximization problem is a novel departure from the traditional epidemic process, not enough motivation is provided. The authors argued that the Ising model allows for multiple flipping of opinions, but the analysis only covered the steady states. The intuition of how the spin system can model the opinion dynamics should be better illustrated with examples. Other related work include epidemic models with flipping states and competing viruses. The mean filed approximation allows for the theoretical development in the paper. However, as the authors pointed out, many Ising model result can be empirically verified and are independent of the numeric techniques. It would much more insightful if the authors can explain the intuitions behind the bifurcation from the opinion dynamics perspective and the phase transition of optimal strategy as interaction strength grows. When analyzing the marketing strategies, the authors compared their algorithm with a few heuristics on the Ising model they started with. The fact that it outperforms others is simply a verification of the approximation. It would be much more convincing if this is done on real influence data that is close to the Ising model. The paper did not quite specify the initial condition of the spin system. h_0 is never formal defined, and how is it differ from the budgeted strategy h? The captions of figures also needs correction. Figure 1 has no center pane, and figure 3 left pane is incorrectly referred to.

Confidence in this Review

2-Confident (read it all; understood it all reasonably well)


Reviewer 3

Summary

The paper proposes an Ising network based model for opinion dynamics and proposes the Ising influence maximization method based on it. Under mean field approximation, it develops several sufficient conditions for when the problem is convex and solvable and a projected gradient descent algorithm. It also establishes several results about the structure of steady-states in the ferromagnetic MF Ising model. It conducts experiments on both synthetic and real world authorship datasets.

Qualitative Assessment

The paper studies the influence maximization using the Ising network model. However, the motivation for using Ising network model for opinion dynamics is not well motivated. Why do we need to use the Ising model? It is less convincing by just saying that this model can capture the steady-state of the opinion dynamics. Some other model can also consider the steady state. For example, how does it compare with prior work such as [1]. In [1], it also considers modeling the steady state of the opinion dynamics. Moreover, how does the discrete-time Ising compare with the continuous-time model in [1]? Some prior work is missing and the paper should have a paragraph clearly stating the stand of this method with respect to other work. As for experiments, it would be good to compare with other standard baselines in terms of the influence maximization. In the experiments it only compares with three heuristics, which is less convincing. Moreover, it would be good to add the scalability experiments on large networks. As for writing, proof sketch could be moved to the appendix. Moreover, the theorems stating sufficient conditions for existence of steady state are less connected to influence maximization, hence makes section 4.1 less connected to other sections. Some terms are also not well defined. For example, what does the susceptibility mean in equation (8)? Any citation on how to obtain equation (5)? Several important literatures are missing: [1]. Learning Opinion Dynamics in Social Networks [2]. Influence maximization in continuous time diffusion networks [3]. Scalable Influence Estimation in Continuous-Time Diffusion Networks [4]. Scalable influence maximization in social networks under the linear threshold model [5]. A data-based approach to social influence maximization

Confidence in this Review

3-Expert (read the paper in detail, know the area, quite certain of my opinion)


Reviewer 4

Summary

In this paper, the authors formulated and studied the Ising influence maximization problem. Under the mean-field approximation, they derived a number of sufficient conditions for when the problem is convex and exactly solvable. In addition, a gradient descent algorithm was presented that efficiently achieves an \epsilon-approximation to the optimal solution.

Qualitative Assessment

- The authors should show how this Ising influence maximization problem captures the characteristics of the influence dynamics in a practical social network; i.e., the authors should justify the model proposed in the paper. - The novelty of this work appears to concentrate on the formulation of the Ising influence maximization. Technical content in Sections 3 and 4 appears rather routine and looks like a standard operating procedure. - The baselines in Section 6 are limited. The network used in Section 6.3 appears to be rather small compared with many large real-world datasets, which may compromise stringency of the results of the numerical simulations.

Confidence in this Review

2-Confident (read it all; understood it all reasonably well)


Reviewer 5

Summary

The paper poses an influence maximization problem as a maximization of the total magnetization in an Ising model on a graph (constrained to a fixed budget of external field). The main analytic result is theorem 6, which shows the existence of a stable non negative steady state. The authors also present an application of PGD algorithm in order to solve this problem.

Qualitative Assessment

The paper is, overall, interesting. My main concern is a "soft" one, and it is about the suitability of using the Ising model for maximizing influence. While it is well studied and it has many nice analytic properties, it is not clear to which realistic scenario it may be applied. The authors give an example of political parties where voting takes place on a later election day - but in this case there are two opposing agents that attempt to maximize the influence. In other words, there is an agent that may add negative magnetic field (a possible extension of the current framework). Alternatively, if there is only one agent that tries to maximize the diffusion of some property into a passive environment, then it is somewhat unreasonable that "non-infected" agents tend to disinfect "infected" agents. For that reason, and many others, it has been somewhat neglected as a model for influence maximization in recent years. Also, there is no analysis of the time scale that on which a steady state is achieved (and what about critical slowdown?) or how to estimate the effective temperature. Both are important in order to apply the suggested model. There is no survey of existing literature and alternative models. The simulation are also somewhat synthetic. Finally, the proof sketchs are unclear. I would remove either one and extend the other. Overall, this is a borderline paper, and I have slight tendency to "accept".

Confidence in this Review

2-Confident (read it all; understood it all reasonably well)


Reviewer 6

Summary

In this paper, the authors propose the problem of Influence Maximization under the Ising network model. Treating the opinion as spin, the Ising Influence Maximization (IIM) problem focuses on maximizing opinion adoption at dynamic equilibrium. The authors provide sufficient conditions that the IIM can be solved under the mean-field approximation via convex optimization. Moreover, the authors provide insights and characterization for the IIM solution which are confirmed via numerical experiments.

Qualitative Assessment

Strength: 1. The paper is technically sound and well written. The definition of the model and the analysis are crystal clear. Moreover, the authors provide interpretation and insights into the theorems which make them very easy to follow. 2. The authors provide interesting and rigorous insights into the IIM problem, including several sufficient conditions where the IIM is solvable under mean-field approximation and characterization of the solution structure. 3. The authors carry out numerical experiments on several synthetic and real networks to support the theoretic analysis. Weakness: 1. The main concern with the paper is the applicability of the model to real-world diffusion process. Though the authors define an interesting problem with elegant solutions, however, it will be great if the authors could provide empirical evidence that the proposed model captures the diffusion phenomena in real-world. 2. Though the IIM problem is defined on the Ising network model, all the analysis is based on the mean-field approximation. Therefore, it will be great if the authors can carry out experiments to show how similar is the mean-field approximation compared to the true distribution via methods such as Gibbs sampling. Detailed Comments: 1. Section 3, Paragraph 1, Line 2, if there there exists -> if there exists.

Confidence in this Review

2-Confident (read it all; understood it all reasonably well)